# Imaging in Gastroparesis: Exploring Innovative Diagnostic Approaches, Symptoms, and Treatment

**DOI:** 10.3390/life13081743

**Published:** 2023-08-14

**Authors:** Francesco Vito Mandarino, Sabrina Gloria Giulia Testoni, Alberto Barchi, Francesco Azzolini, Emanuele Sinagra, Gino Pepe, Arturo Chiti, Silvio Danese

**Affiliations:** 1Department of Gastroenterology and Gastrointestinal Endoscopy, IRCCS San Raffaele Hospital, Vita-Salute San Raffaele University, 20132 Milan, Italy; testoni.sabrinagloriagiulia@hsr.it (S.G.G.T.); barchi.alberto@hsr.it (A.B.); azzolini.francesco@hsr.it (F.A.); danese.silvio@hsr.it (S.D.); 2Gastroenterology & Endoscopy Unit, Fondazione Istituto G. Giglio, Contrada Pietra Pollastra Pisciotto, 90015 Cefalù, Italy; emanuelesinagra83@googlemail.com; 3Department of Nuclear Medicine, IRCCS San Raffaele Hospital, Vita-Salute San Raffaele University, 20132 Milan, Italy; pepe.gino@hsr.it (G.P.); chiti.arturo@hsr.it (A.C.)

**Keywords:** gastroparesis, gastric emptying study, scintigraphy, functional dyspepsia

## Abstract

Gastroparesis (GP) is a chronic disease characterized by upper gastrointestinal symptoms, primarily nausea and vomiting, and delayed gastric emptying (GE), in the absence of mechanical GI obstruction. The underlying pathophysiology of GP remains unclear, but factors contributing to the condition include vagal nerve dysfunction, impaired gastric fundic accommodation, antral hypomotility, gastric dysrhythmias, and pyloric dysfunction. Currently, gastric emptying scintigraphy (GES) is considered the gold standard for GP diagnosis. However, the overall delay in GE weakly correlates with GP symptoms and their severity. Recent research efforts have focused on developing treatments that address the presumed underlying pathophysiological mechanisms of GP, such as pyloric hypertonicity, with Gastric Peroral Endoscopic Myotomy (G-POEM) one of these procedures. New promising diagnostic tools for gastroparesis include wireless motility capsule (WMC), the 13 carbon-GE breath test, high-resolution electrogastrography, and the Endoluminal Functional Lumen Imaging Probe (EndoFLIP). Some of these tools assess alterations beyond GE, such as muscular electrical activity and pyloric tone. These modalities have the potential to characterize the pathophysiology of gastroparesis, identifying patients who may benefit from targeted therapies. The aim of this review is to provide an overview of the current knowledge on diagnostic pathways in GP, with a focus on the association between diagnosis, symptoms, and treatment.

## 1. Introduction

Gastroparesis (GP) is a chronic disorder characterized by gastric dysmotility, resulting in recurrent or persistent upper gastrointestinal (GI) symptoms, such as nausea and vomiting, early satiety, postprandial fullness, bloating, and epigastric/abdominal pain or discomfort, in the absence of a mechanical obstruction [1,2,3]. It significantly impairs quality of life (QoL) in up to 40% of patients, affecting social functioning and mental health [4]. The estimated prevalence of GP is 24.2 per 100,000 persons in the USA [5] and 13.8 per 100,000 persons in the UK [6]. However, data on the epidemiology of GP remain unknown due to symptom overlap with other functional diseases, particularly functional dyspepsia (FD) [7]. Despite a 1.8% likelihood of GP in the general population, the diagnosis rate is only 0.2% [8].

GP has primarily been considered idiopathic (36% of cases), but recent reports have associated it with various comorbidities, most frequently diabetes mellitus (29% of cases), myopathic disorders [9,10], neurological conditions [11], and connective tissue disorders. An autoimmune etiology of GP has recently emerged based on the findings of immune-modulated fibrosis in muscle layers and the loss of enteric nerves [12,13,14] and interstitial cells of Cajal (ICCs) [15] in full-thickness gastric biopsies. Post-infectious etiologies of GP have also been identified [16,17,18]. Additionally, GP can be a consequence of surgical involvement of the vagus nerve, as observed after procedures like vagotomy, esophagectomy, bariatric surgery, or Nissen fundoplication [19].

GP is diagnosed by objectively documenting gastric emptying (GE) delays using gastric emptying scintigraphy (GES) with a standardized test meal to assess gastric retention [20].

The American Neurogastroenterology and Motility Society (ANMS) and the Society of Nuclear Medicine and Molecular Imaging (SNMMI) recommend imaging at multiple time points, including 0, 1, 2, 3, and 4 h after ingestion of a low-fat egg white meal, to enhance the sensitivity of GES [20]. However, the adherence of nuclear medicine laboratories to the guidelines is uncertain [21], and there is ongoing debate regarding the type of test meal and the optimal time point for delayed imaging that should be used to improve the diagnostic yield of GES.

The extent of GE delay in GES has shown a weak association with symptom severity [22,23,24], emphasizing the complex nature of the stomach and the multiple pathophysiological mechanisms contributing to GE impairment. These include vagal nerve dysfunction, impaired fundic accommodation (FA), antral hypomotility, and pyloric dysfunction.

In recent years, new tools for diagnosing GP have been developed, including the 13 carbon-GE breath test and wireless motility capsule (WMC). However, despite their approved use in clinical practice, their role in the diagnostic algorithm of the disease remains unclear.

Meanwhile, new treatments such as Gastric Peroral Endoscopic Myotomy (G-POEM) for refractory gastroparesis have become part of medical practice. Although clinical benefits have been demonstrated, the optimal patient selection for treatment and the GES criteria to validate treatment effectiveness remain subjects of debate.

The aim of this review is to provide an overview of the current knowledge on imaging pathways that can assist physicians in characterizing and managing GP, with a particular focus on the association between imaging features, GP symptoms, and selection and response to treatment. Additionally, this review will elucidate the pathophysiological mechanisms involved in GP, which correlate with imaging findings and have the potential to guide treatment decisions.

## 2. Gastric Neuromuscular Pathophysiology

Despite notable advancements, there are still significant knowledge gaps in understanding the pathophysiological mechanisms of GP [25].

The pathophysiology of idiopathic and diabetic GP involves various cellular changes, such as the loss of ICCs [13], alterations in the enteric nervous system (ENS) and smooth muscle cells [14], and dysregulation in the macrophage population, specifically a reduction in the amount of anti-inflammatory macrophages [26]. These alterations disrupt the intricate balance of gastric neuromuscular function, resulting in impaired motility and delayed emptying. However, further research is needed to better define the specific cellular pathobiology between idiopathic and diabetic etiologies.

The gastric phase of GE initiates once a bolus enters the stomach. During this phase, the bolus is initially retained in the proximal stomach (fundus), referred to as the “lag phase” [27]. Gastric fundus relaxation, which allows for food accommodation, is mediated by vagal innervation from the afferent vagal nerve [25].

Subsequently, the bolus is moved to the antrum for trituration by peristaltic waves.

As the peristaltic waves reach the terminal antrum, pyloric constriction occurs, limiting emptying during the period of peak pressure in the terminal antrum. The contents are forcefully pushed back into the body of the stomach, creating shearing forces that contribute to the trituration of solids, reducing food particle size to ≤2 mm [27]. In this process, antral contractions are mediated by extrinsic vagal innervation and intrinsic cholinergic neurons. Additionally, intrinsic inhibitory mechanisms, such as nitrergic neurons, facilitate gastric peristalsis [25].

ICCs are specialized cells found in the stomach that serve as pacemaker. These cells establish a functional and anatomical connection with fibroblasts and smooth muscle cells through gap junctions and form an electrical syncytium that integrates inhibitory and excitatory neural effects, facilitating the rhythmicity and coordination of gastric contractions towards the antropyloric region [28,29]. The loss or dysfunction of ICCs is a hallmark feature of GP and contributes to the dysregulated motility observed in affected individuals [13,30].

After the bolus is triturated, pyloric relaxation promotes GE.

The relaxation of the pyloric sphincter (inhibition of tone) is primarily mediated by nitrergic neurons and neurons with purine neurotransmitters [31]. Although the concentration of ICCs at the pylorus is decreased compared to the rest of the stomach, the activity of these cells also influences pyloric muscle function [32].

GP is characterized by changes in gastric neuromuscular activity. The identification of such alterations in GP patients holds significant promise for personalized therapy.

Key functional impairments in GP involve alterations in antral motility and, in some cases, pyloric sphincter dysregulation.

Antral dysmotility in GP is characterized by a reduction in the frequency of antral contractions per minute (antral hypomotility) and a decrease in slow wave amplitude and duration [11]. Notably, antral hypomotility appears to correlate with a loss of function or a reduction in the number of ICCs [33]. In post-surgical GP, vagus nerve resection or injury disrupts stomach motility and causes delayed emptying [34].

Pyloric dysfunction is another significant factor contributing to abnormal GE in GP. Hypertonia or spasms of the pyloric sphincter play a role in this dysfunction. Pyloric spasms were initially observed in diabetic GP, suggesting the possible involvement of small fiber diabetic neuropathy in their development [35]. Additionally, damage to the nitric oxide pathway or the loss of ICCs has been associated with dysregulated pyloric relaxation in GP [36].

## 3. Clinical Aspects and Overlap with Other Functional Diseases

GP is a chronic and unremitting disease, with only 28% of cases showing improvement in symptoms [37].

Major GP symptoms include nausea, vomiting, abdominal pain, early satiety, postprandial fullness, stomach distension, and bloating [38]. These symptoms should be evaluated using a validated symptom questionnaire, the Gastroparesis Cardinal Symptom Index (GCSI), in which scores are based on three symptom subscales: (a) nausea and vomiting, (b) fullness and early satiety, (c) bloating and distension [39,40].

Approximately 15% of GP patients experience an acute onset of symptoms [41]. Nausea is the most frequently reported symptom in patients with suspected GP (>95% of cases) requiring clinical evaluation (33% of cases) [42]. The underlying mechanism of nausea in GP is still not fully understood. Functional Magnetic Resonance Imaging (MRI) studies have suggested central nervous system involvement, revealing altered connectivity within the right insula network and bilateral insula network in GP patients after a 30 min exposure to a visual signal inducing motion sickness [43].

Abdominal pain is another prominent symptom that requires evaluation (22% of cases). GP patients with abdominal pain have been found to exhibit elevated somatization scores on the Patient Health Questionnaire (PHQ-15 and PHQ−12, *p* < 0.001), as well as higher levels of depression (*p* < 0.001) and anxiety (*p* = 0.01) [44]. Treating abdominal pain in GP can be challenging [45,46]. Opioids, commonly used in GP for abdominal pain (60% of cases), can complicate the symptom patterns. Opioid users tend to experience worse symptoms, delayed GE, and lower quality of life compared to non-opioid users (*p* ≤ 0.05) [46].

GP symptoms often overlap with those of other functional gastrointestinal (GI) disorders, including chronic unexplained nausea and vomiting syndrome (CUNV) [47], cyclic vomiting syndrome [48], various gut–brain interaction disorders [49,50], and above all functional dyspepsia (FD) [51].

Furthermore, a high prevalence of small bowel dysmotility has been observed in patients with delayed GE [50,52,53]. Constipation is reported in 60% of GP patients, and its severity correlates with worsening GP symptoms, the presence of irritable bowel syndrome, and delays in small bowel, colonic, or whole gut transit. However, the severity of constipation is not associated with gastric retention in GES or WMC testing [54].

In 86% of cases with idiopathic GP, the Rome IV criteria for FD were met [55], particularly for the postprandial distress syndrome. There are ongoing concerns about accurately defining GP, especially in cases where the main symptom associated with delayed GE is epigastric pain. A study conducted by the Gastroparesis Clinical Research Consortium (GpCRC) involving 944 patients with GP (76%) and FD (24%) revealed similarities between GP and FD in terms of clinical presentation, the severity of upper GI symptoms (abdominal pain, nausea, early satiety, and bloating), quality of life scores, and neuropathological findings. Reclassification based on GES results at the 48-week follow-up showed that 42% of initially diagnosed GP patients had normal GE, while 37% of initially defined cases with normal GE showed delayed GE. Based on these findings, the authors proposed that both FD and GP should be considered part of the same spectrum of “organic” gastric neuromuscular disorders [56].

On the other hand, Huang et al. confirmed a significantly longer delay in GE for GP compared to FD (*p* < 0.01). Interestingly, patients with GP-like symptoms and delayed GE had higher Dyspepsia Symptom Severity (DSS) questionnaire scores than those without delayed GE (*p* < 0.01) [57]. The latest guidelines from the UEG and ESNM proposed a distinction between GP and FD based on the attribution of cardinal symptoms. Nausea and vomiting were considered more characteristic of GP, while early satiety, postprandial fullness, and epigastric pain were associated with FD [1].

## 4. Diagnostic Pathways in Gastroparesis

In the presence of GP-like symptoms, upper GI endoscopy should be performed initially to rule out GI mechanical obstruction, malignancy, or strictures caused by peptic ulcer disease. If necessary, a computed tomography (CT) scan should also be conducted [22,58]. GES is considered the gold standard for the assessment of GE, but other diagnostic techniques have been recently developed and used in clinical practice (Table 1) [22].

### 4.1. Gastric Emptying Scintigraphy

GES is a conventional technique that assesses the rate at which the stomach empties its contents into the small intestine.

It involves the administration of a low-fat egg white meal radiolabeled with Technetium-99m.

The imaging process includes acquiring scans with an antero-posterior γ-camera 0, 1, 2, and 4 h after meal ingestion, following the well-established Tougas protocol (Figure 1) [20,59]. The addition of the 4 h timepoint has significantly improved GES diagnostic accuracy, increasing the yield by 50% compared to relying solely on the 2 h timepoint [2]. Research has also shown a positive correlation between GES with a 3 h timepoint and the severity of symptoms in GP patients [60]. Therefore, current ACG guidelines recommend GES for the assessment of meal emptying over at least a 3 h period as the first-line test to diagnose GP in patients with suggestive clinical presentation [22].

The normal values of GE have been established at gastric solid meal retention of >10% 4 h post ingestion; this is considered delayed GE [22]. This quantitative threshold has been found to be reproducible when applied in patients with upper GI symptoms [61]. As mentioned above, GES is considered the gold standard for diagnosing GP.

To ensure accurate GES results, patients are advised to discontinue medications that could interfere with gastric motility for at least two days before the test, including prokinetic, antiemetic, and neuromodulator medications [22].

Despite its value in diagnosing GP, GES has limitations. Standardization of GES procedures remains a challenge across different medical centers due to variations in protocols, equipment, and staff expertise [21]. The use of a low-calorie (255 kcal) and low-fat (2%) egg white meal in GES may not fully represent a typical and normal meal, potentially leading to underdiagnosis of GP. Additionally, radiation exposure raises concerns, especially for women of childbearing age. GES is also time-consuming, and the limited availability of nuclear medicine departments may impact accessibility [62].

In 2008, the ANMS and SNMMI developed a staging classification for GP based on the % 4 h gastric retention values obtained from GES. The classification included four stages: mild (11–20% 4 h retention), moderate (21–35% 4 h retention), severe (36–50% 4 h retention), and very severe (>50% 4 h retention) [20,63]. Subsequently, the ANMS, in collaboration with the American Gastroenterological Association (AGA), proposed an additional classification of GP based on symptom severity, independent of the delay in GE. This clinical classification identified three stages of GP disease: mild (symptoms controlled by diet), moderate (symptoms partially controlled by diet and medications), and gastric failure (uncontrolled symptoms despite conservative treatment) [64].

Over time, many researchers have questioned whether the stages of these two classifications (scintigraphic and clinical) align with each other. In other words, does the severity of delayed global GE correlate with the severity of symptoms? The debate on this topic is ongoing. However, synthesis of the currently available evidence suggests that global GE measurement does not appear to uniquely capture and correlate with GP symptoms.

In a study conducted by the GpCRC in 2017, data from GES were compared with the symptoms of 198 GP patients, revealing that a higher percentage of 4 h gastric retention in GES is associated with more severe early satiety and postprandial fullness [58]. In another study involving 193 patients (including 79 diabetics), it was found that more severe delayed GE specifically correlated with vomiting, rather than early satiety and bloating [65].

Conversely, in the study conducted by Kotani et al., significant differences between diabetic patients with normal GES and those with delayed GES in GI symptoms, including anorexia, nausea, vomiting, abdominal distension, early satiety, heartburn, belching, and epigastric pain, were not found. Interestingly, the authors observed that improvements in GE did not correlate with changes in these symptoms after anti-diabetic medical interventions [66]. Similarly, Kawawura et al. revealed a lack of correlation between GE and GP symptoms in patients with chronic hepatitis C and IFN-induced GP [67].

The relationship between GES and clinical status In GP remains controversial, even after endoscopic treatment. In a single-center French study evaluating the outcomes of G-POEM, the authors found that although there was an overall improvement in the GCSI up to 6 months after the procedure, GES still indicated disturbances in 21% of patients (6 out of 29) [68].

GES provides valuable insights beyond the overall GE measurement by offering information on regional meal distribution, specifically proximal and distal retention. However, studies aiming to correlate localized scintigraphy with symptoms have yielded inconsistent results.

Gonlachanvit et al. investigated patients with FD and Gastroesophageal Reflux Disease (GERD) and found that delayed GE was associated with higher rates of vomiting, nausea, and abdominal distension. They also observed that proximal retention correlated with early satiety [69].

Orthey et al. introduced a parameter called intragastric meal distribution (IMD) based on GES images. IMD represents the ratio of gastric counts in the proximal stomach to the total stomach at any given time, including the initial time (IMD0). Interestingly, they found that low IMD0 values (indicative of impaired FA, <57%) were significantly associated with more severe early satiety but not with other GP symptoms [70].

The meal distribution data obtained during GES hold potential in indirectly identifying the gastric neuromuscular alterations underlying GP, potentially leading to personalized therapy or tailored treatments.

However, the current findings are not conclusive. In a study conducted by Chedid et al., which involved 108 diabetic patients with GP, no significant correlation was observed between IMD0 in GES and gastric accommodation measured by single-photon emission computed tomography (SPECT) [71]. Additional validation is required before considering scintigraphic measurements based on IMD in the assessment of gastric accommodation.

Despite the need for further research, recent evidence suggests optimistic prospects.

In a study conducted by Mandarino et al., a lower median pre-procedural IMD0 value was associated with higher rates of functional response (decrease > 30% in 2 h retention in GES) after G-POEM [72]. A lower IMD0 value indicates antral food retention, likely associated with impaired FA. Therefore, it is reasonable to assume that more distally located gastric disease may benefit the most from endoscopic pyloromyotomy.

### 4.2. Wireless Motility Capsule

The WMC (SmartPill™ Motility Testing System, Medtronic, Dublin, Ireland) is a non-invasive ambulatory test that employs a single-use ingestible capsule to measure transit times throughout the gastrointestinal (GI) tract. Recently, the device has obtained approval from the Food and Drug Administration (FDA) for evaluating GE in patients with suspected GP [22]. The WMC is an indirect test, as the ingested capsule records and transmits pH, pressure, and temperature data from the GI tract to an external recorder. The GE time is calculated based on the rise in pH from the acidic gastric baseline to values above four in the duodenum.

In the study conducted by Kuo et al., a significant correlation (r = 0.73) was found between the GE time measured by the WMC and GE obtained from GES at 4 h in both healthy individuals and patients with gastroparesis [73]. The WMC GE cut-off time of 300 min showed a sensitivity of 65% and a specificity of 87% when compared to GES at 4 h [73]. In a recent study involving 167 GP patients, the WMC detected delayed GE in a higher proportion of subjects (34.6%) compared to GES (24.5%) (*p* = 0.009). The overall agreement in results between the two methods was 75.7% (kappa = 0.42). Notably, in patients without diabetes, the WMC detected a higher proportion of subjects with delayed GE (33.3%) than GES (17.1%) (*p* < 0.001). However, the clinical significance of the higher sensitivity of the WMC in detecting delayed GE remains unclear [74].

Despite these findings, there appears to be little correlation between the detection of GE delay with the WMC and GP symptoms [52].

One major advantage of the WMC is its capacity to assess motility and transit times for the entire gut, including separate assessments of the stomach, small intestine, and large intestine. This presents an exciting option for patients with GP, who often experience constipation issues, as it allows for a comprehensive non-invasive assessment of the entire intestinal transit with a single test.

The primary limitation of the WMC is its inability to correlate the GE value with underlying pathophysiological alterations, such as impaired FA, gastric dysmotility, or pyloric dysfunction. This aspect diminishes its potential as a promising tool for guiding personalized treatment approaches for patients with GP.

Preparation for the WMC test requires discontinuation of medications that may interfere with gastric motility for at least 72 h prior to the test. The test is contraindicated in patients with dysphagia to solid food, swallowing disorders, Crohn’s disease, a history of strictures/fistulae of the GI tract, and abdominal surgery within the past 3 months [75]. The limited availability of the WMC in clinical practice is mainly due to its high cost.

### 4.3. 13 Carbon-Gastric Emptying Breath Test

The 13 carbon-gastric emptying breath test (13C-GEBT) is a non-invasive method used to assess GE for both solids and liquids.

It involves the use of different labeled substances, such as 13C-octanoic acid [76] or 13C-spirulina platensis [77] for solids and 13C-acetate for liquids [78], to track their movement through the GI tract. When ingested, these labeled substances undergo metabolism in the small intestine and liver, leading to the production of 13C-containing metabolites. As these metabolites are released into the bloodstream, they are eventually exhaled through respiration. By analyzing the increase in 13C levels in breath samples using isotope ratio mass spectroscopy [79], researchers and clinicians can indirectly calculate the GE time, providing valuable information about the transit time of ingested substances through the stomach and small intestine.

The GE breath test using 13-carbon spirulina has been validated in simultaneous measurements with the gold standard GES and has shown promising results in both patients with GP symptoms and healthy controls, as well as in pharmacologically induced slowing or acceleration of GE.

In a study conducted by Szarka et al. involving 129 patients with suspected delayed GE and 38 controls, the combination of 45 and 180 min breath samples showed 93% sensitivity in identifying accelerated GE, while the combination of 150 and 180 min samples showed 89% sensitivity for delayed GE [80].

In a study conducted by Viramontes et al., 13C-GEBT detected abnormal emptying with a sensitivity of 86% and specificity of 80% in a cohort of 57 patients, of which 24 had received pharmacological treatment to accelerate or delay GE [81].

Based on these results, the current ACG clinical guideline considers the GE breath test using 13C-spirulina a reliable method to assess GE in patients with suspected GP [22]. The test has also received approval from the FDA [82]. However, in practical terms, 13C-GEBT is still not widely spread in clinical practice.

The advantages of 13-GEBT include the avoidance of elaborate detection equipment and the absence of radiation exposure for the patient. GEBTs can be conveniently conducted on-site, such as in a patient’s office or home, as breath samples remain stable and can be sent to a remote site for analysis. Nonetheless, the test has certain limitations. Firstly, it is an indirect assessment that relies on the measurement of stable isotopes in the breath, which are produced during the metabolism of labeled radioisotopes in the GI tract. This indirect approach may introduce variability and potentially impact the accuracy of the test. Secondly, the metabolism of stable isotopes can be influenced by various factors, such as liver and lung function, which may lead to false results or affect the test’s sensitivity and specificity [79]. Additionally, while the breath test provides information on overall GE, it may not be able to pinpoint the specific underlying mechanisms of GP, limiting its ability to guide targeted treatment strategies.

### 4.4. Other Diagnostic Techniques

High-resolution electrogastrography (HR-EGG) has emerged as a promising non-invasive method for assessing gastric motility in patients with GP. It provides enhanced detection of gastric myoelectrical activity [83].

Studies have shown that patients with GP typically exhibit one- to two-cycles-per-minute patterns and limited three-cycles-per-minute EGG activity compared to healthy controls [84,85]. A prospective international study revealed that patients with CUNV exhibited dysrhythmias in their slow waves. These dysrhythmias included abnormalities in initiation (unstable focal activities and stable ectopic pacemakers) and conduction (wavefront collisions, retrograde propagation, conduction blocks, and re-entry) across different frequencies [86].

Spatial mapping with HR-EGG holds promise in identifying the underlying gastric hypomotility in GP [87]. This represents a promising concept for guiding personalized treatment for patients with GP.

Single-photon emission computed tomography (SPECT) indirectly evaluates gastric tone and is suboptimal in assessing gastric accommodation and sensation simultaneously [88]. However, its widespread use is limited to factors such as ionizing radiation exposure, high cost, and limited availability.

Transabdominal ultrasonography (US) has demonstrated utility and validity in investigating gastric accommodation, GE, and gastroduodenal flow [89].

MRI has shown promise as a diagnostic tool for assessing antroduodenal motility. Hayakawa et al. conducted a study on transplant patients and found a correlation between reduced velocity and prolonged GE using MRI [90].

The hepatobiliary iminodiacetic acid (HIDA) scan is a non-invasive and dynamic study that has the potential to assess GP. Currently, the test has been evaluated in patients with Roux-en-Y gastric bypass [91]. Further evidence will be necessary in the future to validate these investigations in the GP diagnostic pathway.

### 4.5. Endoluminal Functional Lumen Imaging Probe

The Endoluminal Functional Lumen Imaging Probe (Endoflip Impedance Planimetry System, Medtronic, Dublin, Ireland) is an innovative and advanced technology that has revolutionized the evaluation of GI sphincters. This cutting-edge system employs a specially designed endoscopically maneuvered balloon equipped with 16 sensors strategically positioned on a catheter. These sensors are capable of precisely measuring key parameters of sphincters of GI tract, including intraluminal pressure, diameter, Cross-Sectional Area (CSA) and distensibility [92].

In the field of GP, EndoFLIP has been utilized to assess pylorus dysfunction. The measurements of the Pylorus Distensibility Index (P-DI) hold the potential to identify patients who may benefit from pylorus-targeted therapy or verify treatment outcomes.

Extensive research has been conducted using EndoFLIP before or after G-POEM procedures. In a study by Jacques et al., a low pre-therapeutic Pylorus Distensibility Index (P-DI) value (<9.2 mm^2^/mmHg) was found to predict clinical success at 3 months after G-POEM, with a sensitivity of 100% and a specificity of 72% [93]. Another research study showed that a similar P-DI value (<10 mm^2^/mmHg) could predict a symptomatic response to botulinum toxin [94]. However, other studies did not confirm the predictive outcome of pre-operative EndoFLIP measurements for G-POEM procedures [95,96]. However, the study conducted by Vosoughi et al. suggested that post-procedural CSA could be the most reliable predictor for success and the acceleration of GE, with an odds ratio of 1.02 [1.01–1.04] (*p* = 0.008). Specifically, a post-operative CSA greater than 154 mm^2^ with a distension volume of 40 mL was predictive of clinical success at 1 year with a sensitivity of 71% and a specificity of 91% [96].

Further data are required to establish the recommendation of EndoFLIP as a screening procedure in patients with refractory gastroparesis. This would aid in identifying GP individuals with pylorus dysfunction who could potentially benefit from G-POEM interventions.

## 5. Old and New Treatments

Currently, there is a lack of a validated treatment algorithm for GP, and the management of GP is mostly based on a patient-specific stepwise approach. According to the latest guidelines from the AGA, up to 30% of patients have refractory GP, which is defined as “persistent symptoms in the context of objectively confirmed GE delay, despite the use of dietary adjustments and metoclopramide as a first-line therapeutic agent” [22].

### 5.1. Dietary Adjustments

GE is often assessed primarily through GES to evaluate the efficacy of dietary treatment. Two recent randomized controlled trials (RCTs) have shown that a small particle diet appears to improve GE and relieve symptoms in patients with GP who experience predominant abdominal discomfort [97,98]. Therefore, this dietary recommendation should be considered for this phenotype of GP [99]. Other frequently recommended dietary adjustments as part of GP treatment include more frequent meals with low fat and non-digestible fiber content [22], as these can slow down GE. In cases of severe GP, enteral or parenteral nutrition may be necessary.

Eating disorders (EDs), which are present in over 40% of patients with GP symptoms, often meet the criteria for avoidant/restrictive food intake disorder (ARFID). A recent systematic review found no causal association between EDs and GP, and there was no correlation with delayed GE as assessed by GES [100]. However, recent updated guidelines recommend ruling out eating behavioral disorders during the diagnostic work-up of patients with suspected GP and weight loss [22].

### 5.2. Medical Treatment

The primary treatment of GP is focused on improving GE to alleviate symptoms [22]. Dopamine-2 (D2) receptor antagonists have shown significant improvement in both GE (particularly T1/2 in GES when using optimal test methods) and upper GI symptoms [101]. Metoclopramide is the only drug approved by the FDA for the treatment of GP. There is strong evidence supporting its use, demonstrating significant symptomatic relief in various groups of gastroparesis patients. Recent studies have shown that metoclopramide (10/20 mg) provides greater improvement in the total symptom score compared to oral formulations (10 mg) in diabetic patients (type I and II) [102]. An effective metoclopramide trial (at least 10 mg three times a day for at least four weeks) should be administered to patients with GP, although there is not complete agreement in the current guidelines [22]. However, prolonged use of metoclopramide raises concerns about the development of extrapyramidal movement disorders. The estimated prevalence of tardive dyskinesia in metoclopramide users was reported to be in the range of 1–15% [103]. However, more recent data suggest that the prevalence is less than 1% or 0.1% per 1000 patient years [104,105].

Less robust evidence supports the use of domperidone in GP, as it is unable to cross the blood–brain barrier [106]. However, robust data have shown effective improvements in GP symptoms in domperidone users, especially in the diabetic phenotype [107]. It is important to note that this drug has been associated with adverse cardiac events [108]. Both metoclopramide and domperidone have demonstrated significant improvement in GES, particularly T1/2, in real-world settings.

Motilin agonists, such as erythromycin, clarithromycin, and azithromycin, are drugs capable of accelerating GE. According to the most recent network meta-analysis, these drugs appear to be the most efficient in improving GE. However, their administration protocols lasted no longer than 4 weeks and were associated with a significant 15% increase in the risk for myocardial infarction [109].

In recent years, 5-hydroxytryptamine 4 (5-HT4) receptor agonists, which provide serotoninergic prokinetic effects, have gained significant attention in GP drug development. Cisapride was effective in improving antroduodenal hypomotility due to its effect on the MMC, FA, gastric distress sensation, and gastric muscle tone [110]. However, it was withdrawn from the market after evidence of serious cardiac adverse events emerged from a large cohort analysis conducted by the FDA [111].

Prucalopride, the only FDA-approved drug for the treatment of chronic constipation, has also been evaluated for the treatment of GP in a small crossover RCT, showing improvement in GE but not in symptoms [112]. On the other hand, Revexepride did not demonstrate an association with GE improvement in a placebo-controlled double-blind RCT, nor did it show a statistical difference in symptom improvement compared to placebo [113]. In a recent phase IIb RCT, an experimental selective 5-HT4 agonist called Velusetrag showed moderate and dose-dependent effects on GE and symptoms in both diabetic and idiopathic GP patients compared to placebo, without differences between the two phenotypes. However, symptom relief was not sustained in the long term [114]. To date, Felcisetrag has shown the most promising results in improving GE T1/2, 10% small bowel transit, and colonic emptying T1/2 in diabetic and idiopathic GP compared to placebo [115].

Robust data have evaluated the efficacy of ghrelin agonists such as Relamorelin in GP. A recent updated meta-analysis reported significant overall symptom improvement in GP, including early satiety, nausea, vomiting, and abdominal pain (standard mean difference: −0.34; 95% CI: −0.56 to −0.13) [116]. In a phase IIb RCT involving diabetic GP patients, Relamorelin demonstrated a significant improvement in GE and symptoms up to a 12-week follow-up [117].

Ondansetron and Ganisetron, which are 5-HT3 antagonists, showed moderate efficacy in improving GP-related nausea and/or vomiting (76% of patients) for up to 2 weeks, but they did not affect gastric compliance or postprandial accommodation [118,119]. Use of Aprepitant and Tradipitant, both neurokinin-1 (NK-1) receptor antagonists, resulted in acceptable improvements in nausea and vomiting symptoms over a span of 4 weeks [120,121]. However, no correlation with imaging improvement was assessed for NK-1 antagonists. Table 2 shows the main studies on pharmacological treatments of GP.

### 5.3. Surgical and Endoscopic Treatment

Considering that ICC loss, smooth muscle fibrosis, and pyloric spasm unquestionably contribute to the pathogenesis of GP, both surgical and endoscopic pylorus-directed approaches are viable therapeutic options for refractory GP.

In a retrospective large cohort study, laparoscopic pyloroplasty (LP) resulted in GE improvement in over 86% of patients [122]. Other smaller trials reported the overall efficacy of LP in up to 82% of patients, particularly in accelerating GE, as observed in imaging studies [124,125,126]. However, LP is an invasive procedure, and there are no available data on cost-effectiveness analysis in different GP phenotypes. Long-term results indicate that one-third of patients experience relapse [123].

In the case of gastroparesis, as well as in the treatment of other conditions like post-surgical complications, endoscopic techniques have taken precedence over surgery [127,128,129,130,131,132].

Intra-pyloric botulin toxin injection (IBTI) was one of the first endoscopic techniques attempted for the treatment of pyloric hypertonia in GP. After the initial enthusiasm regarding the effectiveness of IBTI in terms of symptom improvement in diabetic GP [133], two recent large placebo RCTs failed to confirm any significant reduction in symptom severity, although some impact on GE was observed [134,135]. Therefore, the latest ACG guidelines do not recommend IBTI [22], partly due to the short-lasting effect of the treatment (average of up to three months) [94,136].

Endoscopic pylorus dilation using a balloon and endoscopic trans-pyloric stenting have been evaluated as potential treatment strategies for refractory GP. The former reported symptom improvement in less than 50% of cases, with the need for further endoscopic re-intervention based on a small retrospective open-label study [137]. The latter demonstrated symptom improvement in 70% of cases of severe GP but with high rates of stent migration (almost 60%) [138,139]. Thus, neither balloon dilation nor trans-pyloric stenting are recommended by the latest ESGE guidelines [140]. Table 3 displays the most significant studies on the surgical and endoscopic treatment of GP.

### 5.4. Gastric Peroral Endoscopic Myotomy

Recently, G-POEM, a procedure involving tunnelization of the gastric submucosa to create a third space for assessing the pylorus followed by pyloric muscle myotomy, has gained increasing interest in endoscopy as a treatment option for refractory GP [141]. After initial experience in ex vivo animal models, the technique has now been standardized and included in the therapeutic approach for refractory GP [140,142]. The technical steps of G-POEM are shown in Figure 2.

In the first large multicenter prospective study involving 30 patients with refractory GP, which were evenly distributed among idiopathic, diabetic, and post-surgical phenotypes, G-POEM achieved a clinical success rate of 86% (follow-up at six months). In this study, GES showed an overall resolution of delayed GE in nearly 50% of patients, although the specific imaging resolution endpoint was not clearly defined (only in terms of post- versus pre-procedural decrease in mean GE T1/2) [143]. A multicenter study, involving 75 patients with a follow-up of up to 12 months, reported a modest clinical response to G-POEM (defined as a reduction in at least one GCSI score with ≥25% decreases in two subscales) in only 56% of patients. A multivariate regression model identified a baseline GCSI score > 2.6 and 4 h gastric retention >20% in GES as the most reliable independent predictors of clinical success (OR 3.23, *p* = 0.04; OR 3.65, *p* = 0.03) [144]. A recent pooled analysis of 10 studies involving 482 patients reported a 12-month clinical response rate to G-POEM of 61% (95% CI: 49 to 71). In this analysis, the distensibility index, measured by EndoFLIP, was significantly higher in the clinical success group compared to the pre-procedural assessment. The rate of adverse events related to G-POEM was approximately 8% (95% CI: 6 to 11) [145].

These percentages of clinical and scintigraphic response have raised reasonable concerns about the actual therapeutic value of G-POEM. However, the first sham-controlled RCT involving 41 GP patients showed the unbiased superiority of G-POEM over the sham procedure at the 6-month follow-up. The clinical success rate, defined as the proportion of patients with at least a 50% decrease in GCSI, was 71% after G-POEM compared to 22% in the sham group (*p* = 0.005). Clinical efficacy was also better in diabetic GP compared to idiopathic and post-surgical GP (89% vs. 67% and 50%, respectively). The median 4 h gastric retention rate in GES decreased from 22% to 12% after G-POEM but did not change in the sham group [149]. This RCT provides the first evidence of the different responses of different GP phenotypes to endoscopic therapy.

Regarding post-surgical GP, given the organic basis of delayed GE, concerns have been raised about the usefulness of G-POEM in this context. A recent meta-analysis reported pooled rates of improvement in GCSI score and 4 h GE delay of 89.6% (95% CI: 72.7 to 96.5) and 81.5% (95% CI: 47.8 to 95.5), respectively [150]. Only one case report highlighted the clinical utility of G-POEM in cases of gastric stenosis resulting from sleeve gastrectomy [151].

G-POEM has demonstrated superiority over other therapeutic approaches for refractory GP. In a retrospective propensity score-matched analysis (23 patients per treatment group, median follow-up 27.7 months), G-POEM exhibited a lower rate of adverse events (26.1% vs. 4.3%) and a longer and more effective clinical response (76.6% vs. 53.7% at 24 months) compared to gastric electric stimulation, particularly in idiopathic GP [146]. Compared to surgical pyloromyotomy, G-POEM achieved a significantly greater reduction in GSCI score and gastric retention in GES at both 2 and 4 h (all *p* < 0.00001), in addition to having lower costs and requiring shorter hospital stays [147].

The optimal selection of patients who may benefit from G-POEM remains a critical issue for its indication and treatment [152] as limited data exist on the correlation between pathophysiological characteristics, imaging, and G-POEM outcomes. As mentioned earlier, diagnostic techniques such as GES [148] and the intra-operative EndoFLIP [93,153] could play a pivotal role in the selection process, but further data are needed.

## 6. Conclusions and Future Directions

GP continues to present significant challenges, profoundly impacting patients’ quality of life and straining the healthcare system. Recent advancements have identified novel approaches for treating GP, with a focus on pylorus-directed interventions like G-POEM. These new treatment techniques have shown symptom improvements in patients affected by GP; however, success rates are still suboptimal.

The current issue lies in the identification of GP patients who can benefit from interventional treatments. Recent studies have found that delayed GE does not significantly correlate with symptoms and does not uniquely identify the physiopathological alterations of the disease, which can unequivocally pinpoint subjects susceptible to endoscopic treatment with G-POEM.

The current impression is that relying solely on GE assessment may not fully capture the complex and multifaceted pathophysiology underlying GP. Experts consider GP as part of the same spectrum of gastric sensorimotor dysfunctions as FD. Recent evidence has shown that measuring GE is not a reliable method for discriminating between GP and FD, as cases of GP can normalize GE over time and many cases of FD can develop delayed GE.

Although GE is still regarded as the “gold standard” diagnostic tool for GP, doubts have been raised about its clinical utility as a biomarker for defining and monitoring GP, leading to the consideration that methods like GES, the WMC, and the 13C-GEBT may play a marginal role in the future management of GP.

Recent research has shed light on other pathophysiological mechanisms beyond abnormal GE that may contribute to GP symptoms. These mechanisms include impaired FA, reduced antral contractions, and alterations in pyloric distensibility. Basic research advances have highlighted the role of altered electrophysiology, such as abnormalities in ICCs and smooth muscle cells.

Exploring these additional underlying mechanisms contributing to the disease represents the first step towards the future of GP management. In this regard, novel diagnostic techniques such as HR-EGG, EndoFLIP, and MRI show great promise in advancing our understanding of GP, aiding in the evaluation of individual patients and guiding the selection of targeted therapies. This represents personalized GP treatment: characterizing the pathophysiology to select the appropriate treatment.

However, despite their potential, these novel diagnostic techniques have not yet achieved widespread clinical use. Some of them are still experimental and require further validation before they can be fully integrated into routine clinical practice. Additionally, the costs and accessibility of these advanced diagnostic tools may present barriers to their widespread adoption. As research continues to unveil their diagnostic capabilities and clinical utility, efforts should be made to streamline their implementation and address any limitations they may currently have.

In conclusion, the lack of specific and objective biomarkers for identifying GP’s pathophysiology and guiding personalized interventions remains a central challenge in the management of the disease. Further research is necessary to validate normative values for each diagnostic technique and explore new strategies for objectively detecting and defining GP. Additionally, more investigation is needed to establish the relationship between measurements obtained through these techniques, GP symptoms, and patient outcomes.

## Figures and Tables

**Figure 1 life-13-01743-f001:**
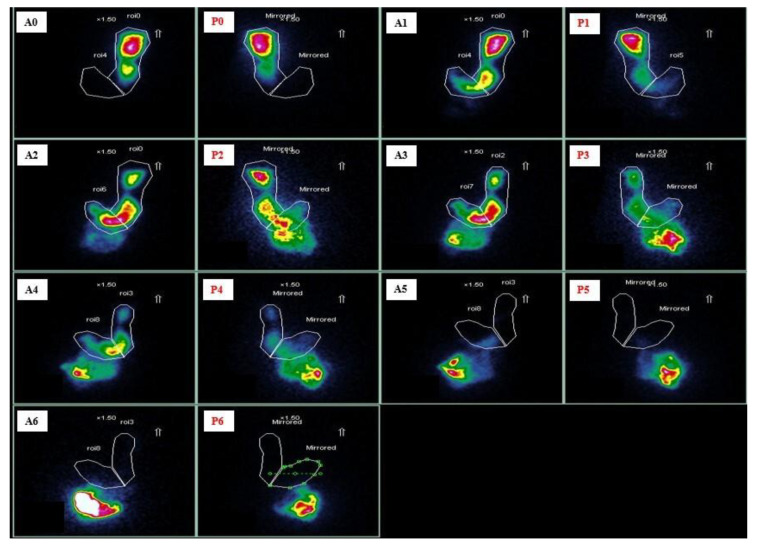
Output of gastric emptying scintigraphy (GES) of a diabetic gastroparesis with primarily antrum dysmotility (intragastric meal distribution of 89%). Anterior rendering at 0 min (**A0**), 30 min (**A1**), 60 min (**A2**), 90 min (**A3**), 120 min (**A4**), 180 min (**A5**), and 240 min (**A6**) after labelled meal ingestion. Posterior rendering at 0 min (**P0**), 30 min (**P1**), 60 min (**P2**), 90 min (**P3**), 120 min (**P4**), 180 min (**P5**), and 240 min (**P6**) after labelled meal ingestion. The copyrights of the pictures belong to the authors.

**Figure 2 life-13-01743-f002:**
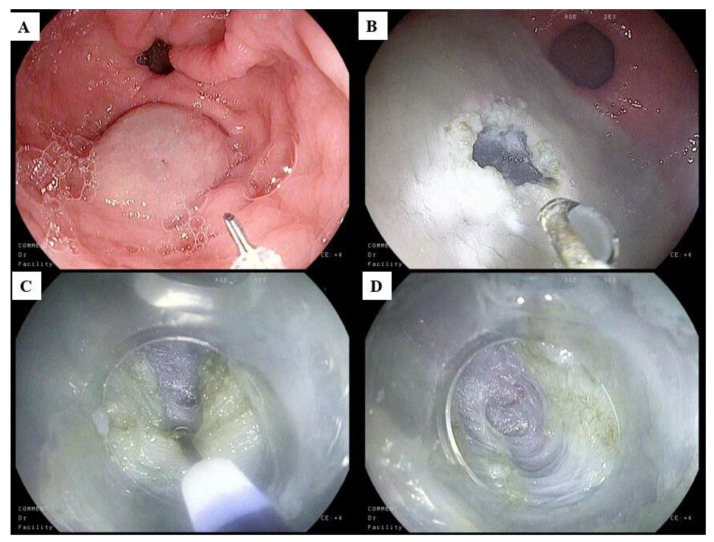
Progressive steps of Gastric Peroral Endoscopic Myotomy (G-POEM): (**A**) the initial lifting by submucosal injection; (**B**) the opening of the tunnel through progressive submucosal dissection; (**C**) the initiation of myotomy from the muscle fibers of the pylorus; (**D**) the completion of full-thickness myotomy. The copyrights of the pictures belong to the authors.

**Table 1 life-13-01743-t001:** Diagnosing tools of gastroparesis.

Diagnostic Technique	Advantages	Disadvantages
Gastric emptying scintigraphy (GES)	The gold standard method to assess gastric emptying Increases diagnostic yield by 50% with the addition of a 4 h timepoint Evaluation of regional dysmotility patterns (IMD and RI) increases diagnostic accuracy	Poor standardization of diagnostic items across different centers Low-calorie and low-fat egg white meal, which does not mimic a normal meal Forbidden for childbearing women due to radiation exposure Low availability of nuclear medicine departments
Wireless motility capsule (WMC)	Non-invasive technique Good performance versus GES Whole gut transit time, including separated evaluations for the stomach and small intestine	High costs Contraindicated for recent abdominal surgery and swallowing disorders
Carbon (13C)-gastric emptying breath test (GEBT)	No requirement of detection No radiation exposure Onsite evaluation with remote analysis	Indirect assessment Liver, lung, and malabsorptive diseases affect accuracy
High-resolution electrogastrography (HR-EGG)	Non-invasive technique Detection of gastric myoelectrical activity	Difficult interpretation of electric signals High costs and low availability
Endoluminal Functional Lumen Imaging Probe (EndoFLIP)	Assesses pylorus integrity	Invasive and time-consuming with high costs No standardized cut-off measures

IMD, intragastric meal distribution; RI, retention index.

**Table 2 life-13-01743-t002:** Studies on pharmacological treatment of gastroparesis.

	Study Design	Patients (n)	GP Subtype	Drug and Posology	Mechanism of Action	Outcomes	Side Effects
Silvers et al. (1998) [112]	RCT Domperidone vs. PBO	286	NA	Domperidone 20 mg QID OS for 4 weeks	D2 receptor antagonist	↓ GCSI from 10.32 to 3.79 in the single masked phase	AEs in 60.1% of patients: diarrhea, headache, abdominal pain, sinusitis, infection
Testoni et al. (1990) [115]	Prospective	20	NA	Cisapride 10 mg QID OS for 15 days	5-HT4 receptor agonist	↓ severity symptoms (*p* = 0.049) and ↑ IDMCs recorded (*p* = 0.022)	NA
Carbone et al. (2019) [117]	RCT Prucalopride vs. PBO	34	Diabetic (n = 6) Idiopathic (n = 28)	Prucalopride 2 mg OS for 4 weeks	5-HT4 receptor agonist	↓ GCSI and GES T½ compared to PBO (1.65 ± 0.19 vs. 2.28 ± 0.2, *p* < 0.0001, and 98 ± 10 vs. 126 ± 13 min, *p*= 0.005).	18 AEs: volvulus (one case), diarrhea (nine cases), headache (eight cases)
Tack et al. (2016) [118]	RCT Revexepride (different dosages) vs. PBO	62	Diabetic (n = 30) Idiopathic (n = 32)	Revexepride 0.02 mg, 0.1 mg, 0.5 mg TID OS for 4 weeks	5-HT4 receptor agonist	↓ GCSI and PAGI-SYM for all dosage groups (*p* < 0.0001); no efficacy difference between drug dosages	102 AEs (43.5% of patients): diarrhea, headache, abdominal pain, dyspepsia, nausea
Kuo et al. (2021) [119]	RCT Velusetrag (different dosages) vs. PBO	34	Diabetic (n = 18) Idiopathic (n = 16)	Velusetrag 5 mg, 15 mg, 30 mg for 12 weeks	5-HT4 receptor agonist	Higher rate of patients with ≥20% T1/2 reduction compared to PBO (52% vs. 5%, *p* = 0.002)	Mild and self-limiting AEs
Chedid et al. (2021) [120]	RCT Felcisetrag (different dosages) vs. PBO	36	Diabetic (n = 11) Idiopathic (n = 25)	Felcisetrag 0.1 mg, 0.2 mg, 1.0 mg IV for 3 days	5-HT4 receptor agonist	↓ mean GES T1/2 in all dosage groups compared to PBO (*p* < 0.001)	Two serious AEs, one discontinuation of the drug due to mild elevated pancreatic enzymes
Camilleri et al. (2017) [122]	RCT Relamoreline (different dosages) vs. placebo	393	Diabetic (n = 393)	Relamoreline 10 μg, 30 μg, or 100 μg TD SC for 12 weeks	GRL receptor agonist	↓ GP symptoms and ↓ mean GES T1/2 in all dosage groups compared to PBO	Three diabetic ketoacidosis and two hyperglycemia events associated with concomitant infections
Carlin et al. (2021) [123]	RCT Tradipitant vs. PBO	152	Diabetic (n = 61) Idiopathic (n = 91)	Tradipitant 85 mg TD OS for 4 weeks	Antagonist of tachykinin receptor 1	↓ nausea compared to PBO; >1 point improvement in GCSI in 46.6% of patients (vs. 23.5% PBO)	31 AEs: diarrhea, nausea, abdominal pain, dizziness, headache

RCT, randomized controlled trial; PBO, placebo; QID, four- times daily; TD, two times daily; OS, oral intake; SC, subcutaneous; IV, intravenous; 5-HT4, 5-hydroxytryptamine; D2, dopamine 2; GRL, ghrelin; IDMC, antroduodenal interdigestive motility cycle; NA, not available; AE, adverse event; GCSI, Gastroparesis Cardinal Symptom Index; PAGI-SYM, Patient Assessment of Gastrointestinal Disorders Symptom Severity Index; GES, gastric emptying study; T1/2, emptying half-time; GP, gastroparesis; ↓ decreased; ↑ increased

**Table 3 life-13-01743-t003:** Studies on surgical and endoscopic treatments of gastroparesis.

	Study Design	Patients (n)	Follow-Up	GP Subtype	Intervention	Outcomes	Adverse Events
Shada et al. (2016) [127]	Retrospective	177	5 years	NA	Pyloroplasty	GP symptoms improvement (*p* < 0.001), except early satiety ↓ post-op median GES T1/2 (pre-op mean 167 min vs. post-op mean 74 min, *p* < 0.001).	Nine AEs: wound infection (four), leaks (two), bleeding (one), pulmonary embolism (one)
Toro et al. (2014) [128]	Retrospective	50	NA	NA	Pyloroplasty	Post-op clinical improvement in 82% of patients ↓ post-op median GES T1/2 (pre-op mean 180 min vs. post-op 60 min, *p* < 0.001)	No intra-operative AEs Five patients (10%) required other GE procedures
Hibbard et al. (2011) [130]	Retrospective	142	3 months	Diabetic (n = 7) Idiopathic (n = 135)	Pyloroplasty	Improvement in all GP symptoms (*p* < 0.001); prokinetic use ↓ from 89% to 14% ↓ post-op median GES T1/2 (pre-op mean 320 min vs. post-op 112 min, *p* = 0.001)	One transient obstruction due to edema Four patients required reinterventions
Friedenberg et al. (2008) [140]	RCT Botulin injection versus PBO	32	4 weeks	NA	Botulin injection	No difference in terms of improvement in symptoms and GE compared to PBO	No complications
Desprez et al. (2019) [142]	Prospective	35	3 months	Diabetic (n = 11) Idiopathic (n = 18) Post-surgical (n = 6)	Botulin injection	Improvement in gastric fullness and bloating in cases with pre-op altered PD (EndoFLIP-assessed) ↓ median TSS from 13.5 to 10.5 (*p* < 0.01)	No complications
Kashab et al. (2015) [144]	Prospective	30	49 days	Diabetic (n = 8) Idiopathic (n = 16) Post-surgical (n = 6)	Trans-pyloric stenting	Clinical response in 75% of patients (mainly in those with nausea and vomiting as predominant symptoms) Post-op GES normalized in six patients and improved in five patients.	Stent migrations in 59% of cases
Kashab et al. (2017) [146]	Prospective	30	5.5 months	Diabetic (n = 11) Idiopathic (n = 7) Post-surgical (n = 12)	G-POEM	Clinical success in 26 patients (86%) Post-op GES normalized in 8/17 (47%) patients and improved in 6/17 (37%) patients	Two minor AEs: one pre-pyloric ulcer, one capno-peritoneum
Vosoghui et al. (2022) [147]	Prospective	80	12 months	Diabetic (n = 19) Idiopathic (n = 33) Post-surgical (n = 28)	G-POEM	Clinical success in 45 patients (56%) GES retention > 20% at 4 h is a predictor of response	Mild AEs in five cases (6%): mucosotomy, capno-peritoneum
Martinek et al. (2022) [148]	RCT G-POEM vs. PBO	41	6 months	Diabetic (n = 17) Idiopathic (n = 11) Post-surgical (n = 13)	G-POEM	Clinical success (decrease in GCSI by at least 50%) for 71% vs. PBO (22%) (*p* = 0.005) ↓ median GES retention at 4 h from 22% to 12%	Ten AEs, only three related to procedures: abdominal pain (one), mucosal injury (one), and delayed dumping syndrome (one)

RCT, randomized controlled trial; GP, gastroparesis; AE, adverse event; GCSI, Gastroparesis Cardinal Symptom Index; GES, gastric emptying scintigraphy; GE, gastric emptying; T1/2, emptying half-time; G-POEM, Gastric Peroral Endoscopic Myotomy; PBO, placebo; PD, pylorus distensibility; TSS, total symptomatic score; ↓ decreased

## Data Availability

Not applicable.

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
