# Peer review of "Imaging in Gastroparesis: Exploring Innovative Diagnostic Approaches, Symptoms, and Treatment"

_life, 2023, doi:10.3390/life13081743_

Round 1

Reviewer 1 Report

The article presents a comprehensive review of the current diagnostic and therapeutic techniques for gastroparesis. Overall, the paper is well-written and will undoubtedly be of great interest to clinicians, patients, and researchers. However, I believe it may be premature to classify high-resolution EGG measurements as a diagnostic tool at this stage. The slow wave characteristics associated with gastroparesis (as discussed in this paper) have predominantly been learned from invasive serosal measurements. Spatial patterns, rather than frequency and temporal components of signals, have been associated with gastroparesis. As such, there is currently insufficient clinical and research evidence to support the idea that body surface EGG can provide such detailed information. I suggest discussing these points and potentially including HR-EGG in Section 3.5 to provide a more balanced view. Despite this minor concern, the paper is certainly a valuable contribution to the field.

Other minor/editorial comments:

Page 1 Line 27: There is an extra period following (EndoFLIP)

Page 2 Line 89: G-POEM should be defined here, but it was done on Page 7.

Page 4 Line 174: FDA should be defined here, but it was done on Page 9.

Page 7 Line 56: AGA needs to be defined.

Page 8 Line 96: Typo in "Reflux"

Page 8 Line 120: though >> through

Excellent work and a very well-written paper.

Author Response

Please, file attached

Reviewer 2 Report

The authors noted to have aimed in this review to provide an overview of the current knowledge on imaging pathways that can assist physicians in characterizing and managing gastroparesis(GP) with a particular focus on the relationship between imaging features, GP symptoms, and selection response to treatment. My concerns are as follows.

1.      There had been so many review articles and expert consensus reports, such as Reference 1, regarding GP so far. Therefore, the authors are recommended to clarify more details of imaging in relation to diverse pathophysiology of GP, that could assist physicians to treat it appropriately.

2.      As mentioned above, complex and various underlying pathophysiology responsible for GP is better to be described, which is obviously crucial in its adequate treatment.

3.      As noted in “Diagnostic pathways in gastroparesis”, gastric emptying scintigraphy(GES) is considered the gold standard for the assessment of gastric emptying(GE). Therefore, more details of techniques evaluating GE function reported in recent literatures, including whole and localized GES, are better to be described in “3.1. Gastric emptying scintigraphy”.

4.      As noted, “13Carbon (13C)-Gastric Emptying Breath Test (GEBT) is another an alternative for assessing GE for solids and liquids” both in adults and children onsite. More details of available findings with GEBT are better to be added in “3.3.13. Carbon (13C)-Gastric Emptying Breath Test” in relation to imaging of GES if possible.

5.      Despite of the aims described in the end of Introduction, any useful descriptions regarding the relationship between imaging features, GP symptoms, and selection and response to treatment are not found, which I consider are crucial in this review. The conclusive messages described in “6. Conclusions and future directions” seem to be much far from the aims of this review and its title.

6.      Although I suspect that the copyrights of pictures in Figures 1 and 2 were belonging to the authors, it is necessary to describe that in figure legends if so.

Nil

Author Response

Please, find attached the response letter

Reviewer 3 Report

The article is interesting and describes significant medical problem in the gastroenterology. To accept the article, some changes should be introduced:

1) In the all manuscript, word "relationship" should be replaced "association"

2) Please create table summarizing pharmacological options for GP contaning drug, mechanism of action, side effects, doses, efficacy, availability etc.

3) Please create table summarizing invasive treatments, including endoscopic and surgical, containing indications, complications, efficacy, availability etc.

4) Please modify the section of conclusion and future directions. Actually, it is too general

Author Response

(The authors gave the same response as above.)

Round 2

Reviewer 2 Report

I truly appreciate the authors' efforts to have revised and upgraded their review article extensively.

Although I am satisfied with the revised version, I just wonder whether the current title "Imaging In Gastroparesis: Exploring the Association Between Diagnostic Approaches, Symptoms and Treatment" is suitable  for the contents. 

Personally, I would like to propose a title "Imaging In Gastroparesis: Exploring Innovative Diagnostic Approaches, Symptoms and Treatment" for this review.

Author Response

Please, find attached the Response letter.

Reviewer 3 Report

I recommend to accept the article in this reviewed version.

Author Response

(The authors gave the same response as above.)
